# A Combined Administration of Testosterone and Arginine Vasopressin Affects Aggressive Behavior in Males

**DOI:** 10.3390/brainsci11121623

**Published:** 2021-12-09

**Authors:** Dilsa Cemre Akkoc Altinok, Mikhail Votinov, Friederike Henzelmann, HanGue Jo, Albrecht Eisert, Ute Habel, Lisa Wagels

**Affiliations:** 1Department of Psychiatry, Psychotherapy and Psychosomatics, University Hospital Aachen, RWTH-Aachen University, 52074 Aachen, Germany; dakkoc@ukaachen.de (D.C.A.A.); mvotinov@ukaachen.de (M.V.); friederikehenzelmann@gmx.de (F.H.); hgjo@kunsan.ac.kr (H.J.); uhabel@ukaachen.de (U.H.); 2JARA Institute Brain Structure-Function Relationship Institute for Neuroscience and Medicine (INM-10), Forschungszentrum Jülich, 52425 Jülich, Germany; 3School of Computer Information and Communication Engineering, College of Engineering, Kunsan National University, 558 Daehak Road, Gunsan 54150, Korea; 4Institute of Clinical Pharmacology, University Hospital Aachen, RWTH-Aachen University, 52074 Aachen, Germany; aeisert@ukaachen.de; 5Hospital Pharmacy, University Hospital Aachen, RWTH-Aachen University, 52074 Aachen, Germany

**Keywords:** testosterone, arginine vasopressin, aggression, Machiavellian traits, inferior frontal gyrus, inferior parietal lobule, fMRI

## Abstract

Aggressive behavior is modulated by many factors, including personality and cognition, as well as endocrine and neural changes. To study the potential effects on the reaction to provocation, which was realized by an ostensible opponent subtracting money from the participant, we administered testosterone (T) and arginine vasopressin (AVP) or a respective placebo (PL). Forty males underwent a functional magnetic resonance imaging session while performing a provocation paradigm. We investigated differential hormone effects and the potential influence of Machiavellian traits on punishment choices (monetary subtractions by the participant) in the paradigm. Participants in the T/AVP group subtracted more money when they were not provoked but showed increased activation in the inferior frontal gyrus and inferior parietal lobule during feedback compared to PL. Higher Machiavellian traits significantly increased punishing behavior independent of provocation only in this group. The pilot study shows that T/AVP affects neural and behavioral responses during a provocation paradigm while personality characteristics, such as Machiavellian trait patterns, specifically interact with hormonal influences (T/AVP) and their effects on behavior.

## 1. Introduction

Historically, aggression can be divided into in reactive (impulsive, emotional) and proactive (instrumental, cold) types [1], while other categorizations have been suggested as well. Today we know that aggressive responses depend on many influencing and regulating factors. This includes (neuro)biological factors, situational variables (e.g., provocation, motivation for reward) and personality traits (e.g., low empathy, hostility). Behaviorally different types of aggression and aggression as assessed in different situations seem to be underpinned by at least partly independent, neural systems [2,3]. Based on previous studies, aggression-related neural pathways underlying the reaction to provocation have been suggested. For example, in a standardized experimental aggression paradigm, the Taylor Aggression Paradigm (TAP), provocation has been associated with increased activity in the rostral ACC, the anterior insula [4], or the mediofrontal gyrus [5]. Aggressive behavior in another widely used experimental task, the Point Subtraction Aggression Paradigm (PSAP), has been positively associated to neural reactivity to provocations by an ostensible opponent in the amygdala, dorsal striatum, insula, and prefrontal brain areas [6]. Although the TAP and the PSAP focus on provocation-related neural pathways, aggression, in these tasks and in other respects, is not only determined by the sensitivity to provocation, but can also be inhibited by controlling behavioral impulses as well. The right inferior frontal gyrus (IFG) in humans has been shown to be critical for behavioral response control [7]. A recent meta-analysis on response inhibition and state anger might be specifically interesting in terms of provocation-related aggression. The right anterior insula and right IFG were shown to be involved in both and seem important for high-order cognitive functioning for aggressive responding [8].

In addition to the neural system, the endocrine system plays a critical role in modulating aggression, likely by preparing the organism to respond differently to external stimuli. Although extensive evidence on the influence of the steroid hormones testosterone (T) and cortisol (C) as well as on their interactions with each other exist in the literature [9], other hormone–hormone interactions have been studied less intensively (but see for example [10]). T and arginine vasopressin (AVP), for example, are both thought to play a role in moderating aggressive responses [11]. However, whether the simultaneous change or interactive change of both hormonal levels may affect humans’ aggression is largely unknown.

In humans, meta-analytical evidence for a direct causal relationship between T and aggression is present but weak [12]. Correlations may depend on individual and contextual factors [13]. For instance, Carré et al. suggested that exogenous T increases aggressive responses exclusively in males expressing high dominance and/or low self-control [14]. Furthermore, T administration could increase or reduce aggressive responding depending on the provocation level, as shown by increased reciprocity to an ostensible opponent’s high or low provocation [15]. Simultaneously to affecting provocation-related aggression, exogenous T in males enhances activation in structures that included medial prefrontal, temporo-parietal and other regions, which showed increased neural activity during punishment selections [16]. In response to social threat, exogenous T has been associated with increased reactivity of the amygdala, hypothalamus, and periaqueductal gray in males [17]. In females, however, it affected the activation of the orbitofrontal cortex (Brodmann area 47) [18]. Despite these associations of neural alterations in response to hormone administration and partly simultaneous observations of behavioral changes, it is important to notice that hormones do not necessarily activate a brain region, thereby initiating a certain behavior. Instead, they might prepare an organism to react to a specific context [19], which then may lead to different neural and behavioral effects.

Similar to T, the neuropeptide AVP, which is released from the posterior pituitary gland, was shown to prepare the organism to react differently in social interactions, including aggression. In hamsters, single injections of AVP increased aggressive behavior [20] and moreover dysfunction of AVP receptors caused reduced aggressive behavior in rodents [21]. One of the first human studies in this field showed that increased cerebrospinal AVP concentration was positively related to a life history of aggression [22]. Correlative findings further suggested that the history of aggression could be associated with higher levels of AVP-related autoantibodies in male prisoners [23]. More substantial evidence than correlative effects may lie in the observation of behavioral changes related to the intranasal administration of AVP. Before a preemptive strike game, in which participants could damage the resources of their opponents, intranasal AVP administration increased the attack rate in both males and females [11].

Conversely, in a social aggression task, AVP had no behavioral effect. However, increased activation in the right superior temporal sulcus during the task was reported in participants who received intranasal AVP [24]. Indeed, across experimental paradigms, AVP administration has been associated with differential brain responses in regions important for social-emotional processes. Based on previous studies, AVP might be able to modulate right amygdala activity in contexts that require empathic behavior [25], activity in the limbic circuitry during social processing [26], activity in the left temporo-parietal junction supporting social recognition [27], and activity in the medial prefrontal cortex–amygdala circuitry involved in emotional regulation [28] as well as reciprocated cooperation [29]. In sum, both AVP and T potentially affect similar brain networks, including prefrontal regions, the amygdala, and the hypothalamus. Thus, both hormones might prepare the organism to react with enhanced aggression depending on the context. However, it is neither clear if actions of T and AVP would be complementary or opposing, nor if they would be interactive or independent.

T and AVP may modulate social behavior via non-genomic processes, thereby acting relatively fast. Findings that point towards a potential mutual (interactive) process are so far limited to different non-human species. T could facilitate aggression via activating AVP receptors in specific brain regions, such as the ventrolateral thalamus [30], and in the medial amygdala [31]. Thus, T and AVP might interactively modulate networks underlying aggression and related social processes. Investigating the potential mutual effects of AVP and T in humans may produce different results: the complementary effects of both hormones would be observed in the form of an increase in the aggressive response. In contrast, opposing effects would dissolve the hormone’s principal action, resulting in a null effect or even reduced aggressive responses. The observation of an increase in aggression could indicate an additive effect by independent hormonal reactions or an interactive effect. Likewise, brain activation that changes in response to AVP and T together may underlie an interaction of both hormones, or independent actions.

Finally, hormones might prepare the organism for a certain response only under certain conditions; for example, in the presence of specific personality traits. In a competition scenario involving monetary reward, Machiavellian traits might be especially relevant. Machiavellian traits refer to manipulative, strategic, and pragmatic features [32], and are associated with high hostility [33]. Individuals with high Machiavellian traits use diverse tactics to reach their aims: on the one hand cooperation and alliance, on the other hand deceit, cheating, lying, revenge, and betrayal [34]. Particularly when they believe it is profitable, individuals with high Machiavellian traits seem to engage in aggression [35]. Machiavellian traits are also defined as the scope of the “Dark Triad” together with psychopathy and narcissism, which have been related to aggressive behavior in the context of relational aggression [36], cyber aggression [37], and reactive aggression [38].

To our knowledge, Machiavellian traits together with hormonal effects have not been studied in the context of aggression directly. Previous results indicated an influence of T on lying [39] and cooperation [40] behaviors that are strongly linked to Machiavellianism. Furthermore, T administration seemed to reduce profit-maximization behaviors in a poker game if this would increase social reputation [41]. Conversely, Pfatteicher et al. showed that only narcissism positively correlated with basal T and C levels, but not Machiavellianism or psychopathy [42]. Another study showed that while narcissism and psychopathy were associated with increased T levels and decreased C levels after a deception task, Machiavellianism was only related to post-test decreased cortisol levels [43]. Although evidence is mixed, Machiavellian traits may be considered as a relevant influence factor in the interaction with hormones.

In the current study, our main goal was to explore aggressive behavior following T and AVP administration using a modified version of the Taylor aggression paradigm (TAP) [15]. We expected a complementary effect (independent or interactive) of the combined hormone administration resulting in enhanced aggressive behavior. Furthermore, we aimed to examine the neural modifications underlying aggressive behavior assessed in an experimental context, studying the AVP/T administration compared to a placebo administration during the presentation of provoking and non-provoking feedback and during the choice of an aggressive/non aggressive response. Based on previous findings regarding the single administration of T or AVP, we expected the combined administration to enhance neural activation in the amygdala, the prefrontal cortex, and the hypothalamus. Since effects of T and AVP on aggressive behavior are likely the result from a combination of multiple traits and state factors, including personality features [44], we further investigated the influencing role of Machiavellian traits. We predicted higher Machiavellian traits to increase aggression, especially after T/AVP administration. Since behavior in the TAP may also vary depending on the belief in the cover story (if the opponent was real), this was investigated as an additional factor.

## 2. Materials and Methods

### 2.1. Participants

The current study included forty-three male participants who were recruited in Aachen via online advertisements and postings. Women were not included because the administration of Testim™ (testosterone gel) (Auxilium Pharmaceuticals, Malvern, PA, USA) is currently limited to males in Germany. Four participants were excluded after the measurement due to heavy movement during scanning, resulting artifacts, or misunderstanding of the task (T/AVP *n* = 19, Placebo *n* = 21).

Participants were included if they met the following criteria: healthy; age above 18 years; no clinically relevant high blood pressure and no current or past prostate tumors; no history of traumatic brain injury and psychiatric or neurological disorders; normal or corrected vision; no contraindications for magnetic resonance imaging (MRI); and right handedness (according to Oldfield) [45]. Groups did not differ with regard to mean age (*p* = 0.570) and verbal intelligence (*p* = 0.996) as measured by the German vocabulary test (Wortschatztest (WST) [46]). Further, the relevant group characteristics (trait aggression, psychopathic personality traits, impulsivity, emotion regulation, Machiavellian traits) did not differ between groups either (see Table 1). Participants received a fixed amount of 50 euros for their participation that could be augmented by the money they won in two further paradigms. Written and informed consent was obtained from all participants in accordance with the recommendations of the Declaration of Helsinki. After the scanning session, participants were fully debriefed about the study aims and the cover story around the paradigm. The Ethics Committee of the Medical Faculty of the RWTH Aachen University approved the procedures of the experiment.

Our study design did not include groups in which only a single hormone was administered. As a preliminary attempt to disentangle the effects of the combined T/AVP administration from an exclusive administration of T, we selected a comparison group from a previous study [16] with the same design and compared the retrospectively matched sample with the groups from this study. In addition to the currently presented two groups, we here included nineteen males who had received the same dose of T administered. The Appendix A includes the details about the selection of participants, sample characteristics, and procedures of the additional data analyses.

### 2.2. Procedure

This study was a follow-up pilot study of a larger placebo-controlled, randomized, double-blind study testing the effect of exogenous T administration and its neurocognitive modulation during aggression and risk-taking [15,16]. A detailed description about the complete study procedures can be found in the previous publication (Figure 1).

The total duration of the study was about six hours and started between noon and 2:00 p.m. in order to reach a stable circadian baseline hormone level. To improve the credibility of the cover story for the TAP, an ostensible male opponent was introduced to the participant, who would compete in a reaction time task in a separate room with a computer linked to the scanner. After taking a first blood sample (T1) to determine the baseline serum levels, participants received either a transcutaneous 5 g Testim^TM^ gel corresponding to 50 mg T on their shoulder or an equivalent amount of sonography gel (placebo). After participants filled in personality questionnaires, they completed several short tasks (indicating the preferred personal distance towards visual stimuli and performing a frustration task), provided saliva samples for genotyping [52], and had about 1 h break before the scanning session. In order to detect the changes in T levels, another blood sample (T2) was collected before the administration of AVP. Intranasal AVP (250 μL, 20 IU in 0.9% NaCl, Sigma, Germany) was administered (equivalent to 0.025 IU intravenous hormone application) to subjects who received T at the beginning, while the other participants received a placebo spray (identical intranasal administration). Previously, it has been demonstrated that intranasal versus intravenous AVP administration directly influences the neural level [53]. Intranasal AVP administration was used effectively many times [54,55,56], also together with oxytocin [57]. The fMRI session started immediately after AVP/PL administration. In the fMRI sessions, the first task was the TAP, followed by a risk-taking task (BART), not included here. After scanning, a final blood sample (T3) was collected. We debriefed participants after assessing if they believed they belong to the experimental or the placebo group.

### 2.3. Hormonal Levels

We determined T in blood serum samples. The samples were analyzed with immunologic in vitro quantitative determination of T in human serum and plasma (Electrochemiluminescence immunoassay, ECLIA; Roche Diagnostics GmbH, Mannheim, Germany). In order to verify the treatment success on T levels, a repeated-measures ANOVA was performed with time as within-subject variable and treatment group as between-subject variable. AVP was not analyzed as the blood serum levels likely do not correspond to cerebrospinal fluid (CSF) levels as the blood–brain barrier limits the flow [58].

### 2.4. Modified Taylor Aggression Paradigm (TAP)

In order to study aggressive behavior in reaction to social provocation, researchers developed different provocation paradigms. The TAP is one of the first paradigms assessing reactive aggression through provocation [59]. The original paradigm was created as a competitive situation, whereby a subject and a fictitious opponent can provoke and retaliate against each other by administering electrical shocks after winning a reaction time task. Therefore, the TAP is also frequently referred to as a “competitive reaction time task” and different modern versions of such aggression paradigms have been suggested to be a powerful tool in aggression research [60]. In the literature, there are multiple modified versions of the paradigm with different modalities that have been shown to successfully induce and measure aggression [61]. Recent analyses suggest that the task behavior (e.g., retaliation) can be ascribed to the preceding provocation and the success during the reaction time task [15,61,62]. In the current version, a monetary reduction is used as provoking stimulus as well as an indicator of aggressive behavior. Before each trial, participants determined the amount of money (between 0 and 100 cents) that they wish to take from the opponent in case the opponent will lose. They could select the amount of money on a visual analogue scale. For the subsequent reaction time task, participants were asked to react as fast as possible to an appearing soccer ball moving across the screen. Following the reaction time game, participants received feedback regarding the trial outcome (win or loss) and, in case of losses, the money subtracted by the ostensible opponent. This served as a provocation, while winning trials, in which participants did not see the decision of the opponent, served as control (no provocation). These different conditions were analyzed as contrasting conditions for the behavioral and fMRI data. Participants did not earn the money they decided to subtract. They only won 50 cents in the win trials. Overall, the paradigm lasted 25 min. The TAP consists of 54 predefined lost trials (23 high provocation trials: 80–100 cents; 25 low provocation trials: 0–20 cents; and 6 medium provocation trials: 30–70 cents) and 30 trials in which participants won. Minor variations could emerge when individuals’ reaction time was above 600 ms. In these cases, individuals lost the trial, followed by a medium provocation trial (50 cents). Participants were asked if they believed that the opponent was a real player after the TAP.

### 2.5. Questionnaires for Neuropsychological Assessment

We administered a German version of the Machiavellianism Scale (MACH-IV) questionnaire, consisting of 20 items, to measure Machiavellian traits [51]. The scale has four subscales (negative tactics, positive tactics, cynical view, and positive view). High scores on this scale indicate more manipulative behaviors, low ability to emotional connections with others, and reduced recognition of common morality rules.

### 2.6. fMRI Data Acquisition

Imaging data were collected using a Siemens 3 Tesla Prisma scanner (Siemens AG; Erlangen, Germany) equipped with a 12-channel head matrix coil and located in the Department of Psychiatry, Psychotherapy and Psychosomatics, RWTH Aachen University. To restrict movements, we used foam paddings. A time series of about 745 functional images per participant was acquired. We used a spin-echo EPI sequence with the following acquisition parameters: TR = 2000 ms; TE = 3 ms; flip angle = 77°; FOV = 192 × 192 mm^2^; matrix size = 64 × 64, 36 slices; voxel size = 3 × 3 × 3 mm^3^; and slice gap 0.8 mm. Functional scans lasted 25–30 min. Structural scans were acquired using a T1-weighted MPRAGE sequence with the following acquisition parameters: TR = 2300 ms; TE = 2.98 ms; flip angle = 9°; voxel size = 1 mm^3^; and inter-leaved, distance factor: 50%.

### 2.7. Data Analysis

All behavioral data analyses were performed using IBM SPSS Statistics 25 while fMRI data were analyzed with the SPM12 toolbox implemented in MATLAB 2018.

#### 2.7.1. Hormonal Levels

T levels were analyzed with immunologic in vitro quantitative determination of T in human serum and plasma (Electrochemiluminescence immunoassay, ECLIA; Roche Diagnostics GmbH, Mannheim, Germany). Hormonal levels of the T/AVP and placebo group (PL) were compared at each time point to check if there were changes in T levels of the T/AVP but not the PL group. For this purpose, a repeated-measures ANOVA was performed with time (T1 = before administration, T2 = 3.5 h after testosterone administration) as the within-subject variable and administration group (T/AVP, PL) and belief (hormones, no hormones) as the between-subject variables. Since due to measurements problems the hormone data for T3 (after the task) existed only for 20 individuals, we present an additional analysis, including this time point (see Appendix A). In the Appendix A, we also present an additional analysis of hormonal levels, including an age-matched group, which received only testosterone gel (for details, see Appendix A). The AVP levels were not analyzed due to the limited evidence that serum levels relate to AVP in CSF.

#### 2.7.2. Taylor Aggression Paradigm (TAP)

Behavior in the TAP was investigated by applying a repeated-measures ANOVA with a within-subject factor, provocation (win = no provocation, loss = provocation), and two between-subject factors, administration group (T/AVP, PL) and belief (hormones, no hormone). The dependent variable was the mean amount of money participants subtracted in the subsequent trial following either a win or a loss (scale: 0–100 Cent). Again, we performed an additional analysis on the TAP, including an age-matched comparison group that received testosterone administration only (for details, see Appendix A).

#### 2.7.3. fMRI

Each individual time-series was fitted to a general linear model (GLM) creating a task-specific model (first level). In total, we modelled several regressors as categorical variables: feedback phases of lost trials were modelled separately for no provocation (win) and provocation (loss) and an additional regressor encompassed win trials. The decision phases were modelled separately for trials after no provocation (win) and provocation (loss) equally. Finally, all game phases were modelled with a single regressor. Jitter and inter-trial intervals were not modelled, thus serving as an implicit baseline. The stimulus functions were convolved with the hemodynamic response function. In addition, realignment parameters (three rigid-body translations and three rotations) as well as an intercept for the complete scanning session were added. A high-pass filter (128 s) was applied to remove low-frequency drifts. Parameter estimates for all regressors were obtained after accounting for temporal autocorrelations (AR1).

At the group level, two full factorials were set up with a substance group (T/AVP, PL) as a between-subject factor and condition (loss/win) as a within-subject factor. In the first model, we tested the BOLD signal differences during the decision period (after loss/win); in the second model, we tested the BOLD signal differences during the feedback period (upon loss/win) at a cluster defining threshold of *p* < 0.001 and a family-wise error cluster level threshold of *p* < 0.05.

Parallel to the behavioral and hormone level analyses, we performed an additional analysis with the same design, but including a three-level administration factor (T/AVP, PL, T) that contained an age-matched group that received testosterone administration only (see Appendix A for details).

#### 2.7.4. Correlation Analyses

Pearson correlations were calculated between the MACH-IV and the total amount of money subtractions in response to provocation (loss) and no provocation (win) trials to determine the relationship between Machiavellian traits and aggressive responding. Correlations were performed separately for each group (T/AVP and PL). At the neuronal level, we tested if the total score of the MACH-IV scale with the brain signal in which the treatment effect was observed by extracting the mean signal from the respective region.

## 3. Results

### 3.1. Participants

To check whether participants noticed which substance they received, all participants were interviewed after the experiment. Their beliefs were not significantly related to the actual substance they received (Pearson Chi-Square, X2(1, *n* = 39) = 1.035, *p* > 0.05).

### 3.2. Hormone Concentration

A repeated measures ANOVA showed no main effect of group, F(1, 36) = 2.91, *p* = 0.096, or time, F(1, 36) = 3.14, *p* = 0.085. However, the interaction of administration group and time (Figure 2) was significant, F(1, 36) = 19.68, *p* < 0.001. At T1, the PL group and the T/AVP group did not differ, F(1, 36) = 0.31, *p* = 0.584 but groups differed significantly at T2, F(1, 36) = 10.23, *p* = 0.003. In the PL group, there was no significant change, F(1, 36) = 3.31, *p* = 0.077, but in the T/AVP group, we observed a significant increase of testosterone levels from T1 to T2, F(1, 36) = 20.81, *p* < 0.001 (Figure 2). The belief about being in the hormone or no hormone group had no significant effect, F(1, 36) = 3.68, *p* = 0.063, and there was no interaction with time, F(1, 36) = 2. 68, *p* = 0.110, or real administration group, F(1, 36) = 0.025, *p* = 0.619. There was no three-way interaction either, F(1, 36) = 2.28, *p* = 0.140.

### 3.3. Task Behavior

The main effects and interactions are presented in Table 2.

The interaction of administration group (T/AVP, PL) and provocation condition (loss, win) was significant (Table 2, Figure 3) and thus post hoc tests were calculated. Comparing the provocation conditions within groups showed no difference in the PL group, F(1, 36) = 0.35, *p* = 0.561 between either won trials (M = 45.22, SE = 7.44) or lost trials (M = 45.78, SE = 7.41). However, money subtractions were significantly higher after won trials (M = 52.51, SE = 6.91) than after lost trials (M = 49.75, SE = 6.88), in the T/AVP group, F(1, 36) = 9.60, *p* = 0.004. Groups did not differ in the amount of money they subtracted after won trials, F(1, 36) = 0.50, *p* = 0.824 or after lost trials, F(1, 36) = 0.20, *p* = 0.887. The main effect of belief, F(1, 36) = 2.34, *p* = 0.135 was not significant. Moreover, the belief did not interact with the actual administration group, F(1, 36) = 0.005, *p* = 0.944, and not with provocation, F(1, 36) = 0.624, *p* = 0.435, or provocation and administration group, F(1, 36) = 0.99, *p* = 0.755.

Correlations of the total score of the MACH-IV scale with monetary subtractions after winning (no provocation) or losing (provocation) are depicted in Figure 4. In the T/AVP group there was a positive correlation with the MACH-IV score and subtractions after winning, *r* = 0.588, *p* = 0.008, or losing a trial, *r* = 0.569, *p* = 0.011. In the PL groups there were no correlations between monetary subtractions and the MACH-IV scale (loss: *r* = −0.062, *p* = 0.797; win: *r* = −0.065, *p* = 0.786).

### 3.4. fMRI Results

To investigate the effect of provocation on neural activation, no provocation trials (win) and provocation trials (loss) were compared. Losses mainly led to convergent activation in visual areas, while winning was reflected in regions in the prefrontal and medial cortex as well as the limbic system and the basal ganglia (see Figure 5). The main contrasts of won versus lost trials in the feedback period, when participants saw that they won 50 cents or lost and the amount, the other player subtracted (Table 3), and the following decision period, when participants could select what they would like to subtract in turn (Table 4), revealed significant results in both directions.

Group differences were observed in the feedback period only and they were observed across the win and loss trials (see Figure 6). In the T/AVP group we observed increased activation in the left inferior frontal gyrus (IFG) extending to the left insula and the left temporal pole (peak at x = −52, y = 6, z = 12; t(74) = 4.73, *p* = 0.001). A second cluster showing increased activation was in the left parietal lobule (peak at x = −50, y = −44, z = 58; t(74) = 4.72, *p* = 0.031). No significant interaction of the provocation and administration group was observed.

An additional model testing the influence of belief by independent samples t-test showed no significant effect on a whole-brain level. Similarly, we did not observe any significant correlation with MACH-IV (*r* = 0.285, *p* = 0.224) correlating the extracted mean signal from the main effect of the hormone group in the feedback period.

## 4. Discussion

The present study investigated hormonal modulation of aggression and the additional influence of Machiavellian traits. On the neural level, we found increased brain activation across different cortical and subcortical regions when participants were provoked. This is in line with previous experiments that administered the TAP [62,63,64]. In the hormonal administration group, a cluster including the left insula and inferior frontal gyrus (IFG) as well as the left parietal lobule exhibited a stronger neural response during the feedback phase. This activation was independent of winning or losing a trial and thus independent of the presence of provocation. Therefore, the observed group difference in the neural signal does not relate to reactive aggression. Instead, the hormonal administration seems to prepare different neural activity in response to the overall contest. Similarly, hormone administration had only weak effects on the behavioral level, which did not point towards an enhanced reaction to provocation. In the T/AVP group, participants seemed to subtract even more money if they had won a trial, which opposed our hypothesis. This behavior was not observed in the PL group. Most interestingly, however, in the T/AVP group monetary punishment during the TAP was higher if participants had high Machiavellian traits while it was lower in those with low Machiavellian traits. This relationship was independent of provocation in the T/AVP group. In contrast, it was not present in the PL group, neither after provocation nor non-provocation trials.

### 4.1. Hormonal Modulation of Aggression

Hormonal levels of T increased in the T/AVP and T group in contrast to the PL group, in which T levels did not change. This corroborated a successful manipulation of T levels. At the same time, there was no indication for an influence of the AVP administration on T levels, as noted in an additional comparison to a group that received testosterone only but underwent the same procedure as the T/AVP group. AVP administration thus did not seem to affect T levels in blood serum, a result that has to be replicated in a larger sample.

During the TAP, the main result showed that participants in the T/AVP group subtracted more money after they won a trial. Critically observed, in won trials, participants were not provoked, since their reward was presented as feedback. Thus, aggressive behavior after these trials could either reflect a retaliation to previous trials or more proactive aggression. We can only speculate here, and the finding clearly did not support the original hypothesis of increased reactive aggression. Our previous study reported increased responsiveness towards provocation when investigating the effect of T administration in the TAP [15]. We thus cautiously assume that in combination with AVP, the administration of T seems to influence decisions differently. However, it should be noted that, in the literature, the effects of T administration on aggressive behavior are not consistent and might vary due to other influence factors. T administration was, for example, shown to increase aggressive behavior during the PSAP [65], but in another study, this was only observed in participants with increased impulsivity [14]. T also elevated responsivity in a modified ultimatum game [66] but did not influence the implicit aggressive response in a non-social context [67]. It did not promote human aggression after a prolonged period of administration either [68]. Therefore, we suggest that the testosterone manipulation influences the intensity of the aggressive response only in a given/specific context or if other influence factors support the hormonal effect. AVP may be one of these factors, but this has to be replicated in larger, fully randomized controlled studies.

### 4.2. Hormonal Modulation of Brain Signals

In combination with AVP, T administration seemed to influence specific neural networks. In terms of neuronal responses, the combined hormone administration increased activation in the left inferior frontal gyrus (IFG), extending to the left insula and the left inferior parietal lobule (IPL). Interestingly, the IPL has been associated with economic decision-making when fairness evaluations are unclear [69]. In situations of low cooperation, the activity in this region has been noted to increase [70]. On a larger scale, the IPL may be allocated to the mentalizing network [71,72], which supports the understanding of other individuals’ mental states. Although it is not possible to infer a specific cognition based on signal changes in the human brain, we suggest that the hormonal administration could have influenced cognitive states that determine the evaluation of other’s behavior. Assuming that T and AVP primarily affect networks involved in social cognition, this could also explain, why the observed outcome (increased or decreased aggression) in response to the hormone administration strongly depends on the context or study design. Furthermore, personality traits may influence how a situation is perceived and signal alterations in social cognition networks may therefore have different effects on individuals.

Another cluster that showed increased activity in the T/AVP group was the IFG (including anterior insula). Increased IFG activity is associated with many different cognitive and emotion-related processes such as emotion regulation [73], empathy processing [74], and evaluation of emotional cues [75]. Since there are many different cognitive and emotion related processes associated with IFG reactivity, the role of the IFG in the current experiment is unclear. One possibility may be that the IFG modulation may influence the processing of monetary wins and losses [76]. Another important function would be inhibitory control and response inhibition [77,78,79], which, in the current task, is important for individuals to successfully inhibit their impulse to subtract money from the opponent. Albeit the administration effect was not observed during the decision period, but the feedback period, in which no response inhibition is required. Finally, the IFG may also be involved in mentalizing processes [80]. There is even evidence for the IFG and the IPL working together in a feedback loop, as suggested by a study on divergent thinking [81]. Possibly, signal changes in the IFG and IPL might influence the evaluation of the fairness of the opponent’s behavior, which could further determine if the participant perceives it as provocation or not. Hormone-induced changes in these regions that largely influence social cognition and emotion may thus lead to the observed small behavioral differences between groups. However, the behavior was affected in response to win trials only, while on the neural level the effect was observed across all trials. Thus, relating behavioral and neural changes in response to hormonal administration must be reconsidered and both effects may be independent. While in the current study we primarily observed an influence of AVP/T administration on the left IFG, extending to the left insula and the left IPL, we cannot exclude that the neuromodulators may affect behavioral output by modulating other brain regions on a subthreshold level. A previous study, for example, pointed out that activation of AVP1b receptors, which are specifically present in the hippocampal CA2 region, may affect social forms of aggression [82]. In particular, the authors suggest that this receptor activation may modulate the hypothalamic–pituitary–adrenal axis under acute stress. In the current study, we may not have observed changes in the activation of the hippocampus because social provocation is only a mild social stressor and may not have led to an acute stress response. Moreover, in rats it has been shown that a vasopressin receptor antagonist can suppress the distributed neural circuit involved in aggressive motivation by specifically activating the anterior thalamic nuclei [83]. It currently remains unclear if such findings may be translated to humans mostly due to differences in the definition of the neural circuits, which in humans may include a broad involvement of prefrontal brain regions [84].

Finally, instead of a direct causal link of T and AVP and aggression, there are many adjusting screws that may influence if aggressive action comes into play. Referring to animal studies, external manipulation of the neurotransmitter systems (e.g., serotonin, GABA) may ultimately activate brain systems related to stressful situations [85] or aggressive behavior [21,86]. Moreover, genes might interact with hormones, modulating behavior only in certain groups [16,52].

Overall, the results regarding neural alterations in the T/AVP group either point to an interactive or independent effect of T and AVP on the same brain regions. However, due to the study’s pilot character, we emphasize that all conclusions are preliminary and need to be investigated in a fully randomized controlled investigation with larger group sizes. Furthermore, only studies that test both, the effect of the combined and the single hormone administration might clarify if the modulatory effect of T and AVP is specific for the combination or if the observation of the signal modulation in the IPL and IFG is mainly driven by one hormone. Our preliminary comparison supports the specificity of the combined administration as we observed a similar neural pattern as comparison between the AVT/T and the T group as in the main comparison of the AVT/T and PL group.

### 4.3. Machiavellian Traits

One of the modulating factors that seemed to be relevant for determining aggressive behavior are Machiavellian traits. As shown in this study, positive associations of Machiavellian traits and aggression levels in the TAP were observed in participants who received the combined hormonal administration. This association was not observed in the PL group. Interestingly, individuals with high Machiavellian traits (high MACHs) subtracted more money, independent of provocation. They maintained their higher monetary punishment level throughout the task, but only if they had received hormonal administration. These results certainly have to be interpreted with caution as we did not specify a hypothesis on this, and the analysis might be susceptible to statistical errors (false positive). However, as discussed and shown before, hormone effects may actually depend on personality traits such as impulsivity, dominance, and self-control. These traits have already been shown to modulate T effects on aggression [14]. Moreover, assertive high power has previously motivated a greater T increase during a social dominance contest [87] and these T changes were associated with aggressive behavior in individuals with an independent self-construal [88] or high values in grandiose narcissism [89]. Our results contribute to the literature, emphasizing the complex effects of aggression on social and personality features, and neuroendocrine mechanisms. Enhanced activation of neural social cognition systems via T and AVP may increase social cognition, but the interpretation of the situation and the resulting decision might still largely depend on personality traits.

High MACHs are known for their manipulative, strategic, pragmatic strategies, and deceptive intentions to reach their goals [34]. Previous findings show a strong relationship between proactive aggression, manipulative features, and Machiavellian egocentricity [90]. Ambitious financial attitudes of high MACHs have been widely discussed in the literature [91]. Especially the focus on financial reward or loss in this TAP version may thus have affected individuals with high MACH traits. High MACHs hold the view that the end justifies the means [92]. Ultimately, this might be a high motivator in particular for unprovoked aggression (after winning money), which was increased in high MACHs under hormonal treatment.

### 4.4. Limitations

There are several limitations in the present research. We probed the influence of a combined administration of T and AVP on aggressive behavior. However, while previous studies investigated the single administration of T and AVP, this study misses randomized control groups that received only T or only AVP. Therefore, conclusions about the single influence of the respective hormone are speculative and the results need to be confirmed in a fully randomized controlled trial. In addition, while we measured plasma T levels, we did not assess AVP concentrations in CSF to test the success of hormonal manipulation since CSF sampling from human subjects is painful. In the literature, it is shown that intranasal AVP administration increases the CSF AVP concentration; however, plasma levels are not affected [93]. Therefore, plasma levels were not measured. Future research may use other approximations to estimate the AVP level changes, such as measuring copeptin [94]. Consistent with most prior research, the current study investigated a homogenous sample of males, which certainly limits the interpretation. There is a high need for studies that include larger groups of males and females and observe participants of a larger age span. Eliciting aggression in the MRI environment with limited spatial and temporal freedom limits the ecological validity of the TAP. Although we adjusted our study design to acknowledge these limitations from prior studies, some of them remain because of the difficulties to create a natural setting. Finally, future research should examine additional personality factors, which may be influenced by hormone administration.

## 5. Conclusions

The present study gives insight into the hormonal influence, personality factors, and neural mechanisms on aggressive behavior. The combined administration of T/AVP may induce a reduction of aggressive tendencies but may also increase aggression in participants with high Machiavellian traits. Behavioral changes may be facilitated via enhanced activation of brain regions relevant for social cognition processes. We suggest that complex behavior, such as reactive aggression, is influenced by an interplay of different connected factors, such as situational changes, personality traits, as well as attitudes and believes.

## Figures and Tables

**Figure 1 brainsci-11-01623-f001:**
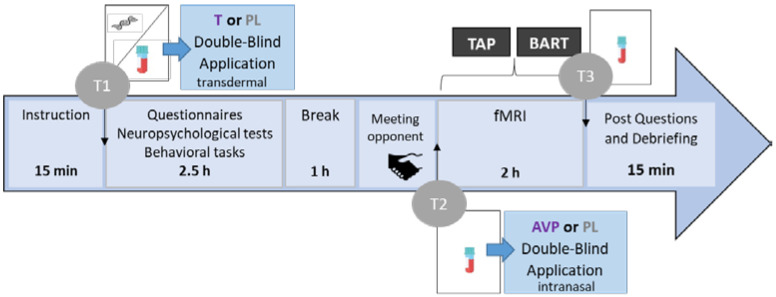
The figure presents an overview of the course of the complete study procedure. T; testosterone, AVP; arginine vasopressin, PL; placebo, TAP; Taylor Aggression Paradigm; fMRI; functional magnetic resonance imaging; T1; timing of first blood sample; T2; timing of second blood sample; T3; timing of third blood sample.

**Figure 2 brainsci-11-01623-f002:**
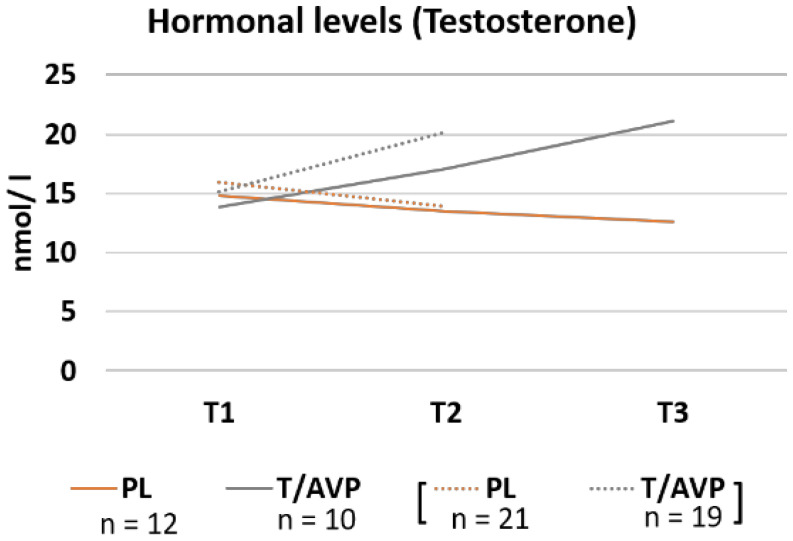
For visual purposes, the mean testosterone raw levels are presented for the placebo (PL) and the arginine vasopressin and testosterone group (T/AVP) for baseline measures prior to administration (T1), pre-task measures 3.5 h after administration (T2), and post-task measure (T3). Note: Groups were smaller at T3 due to data loss. Full data available at T1 and T2 are presented in dotted lines while data on all participants that were measured three times is presented in solid lines for T1, T2 and T3.

**Figure 3 brainsci-11-01623-f003:**
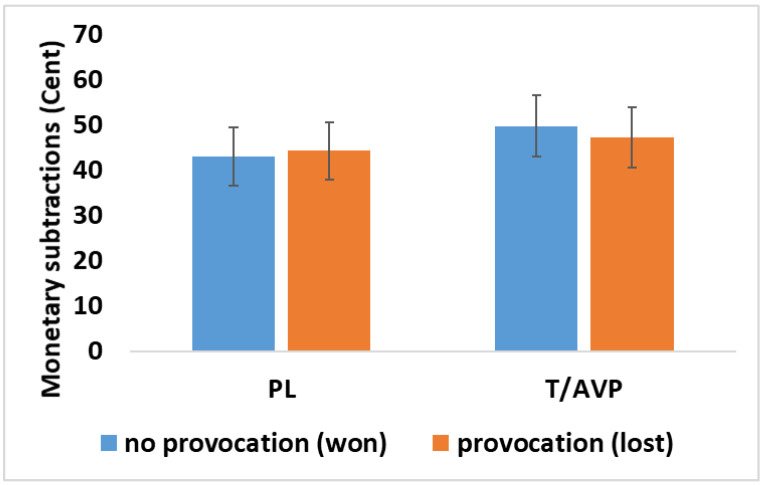
For visual purposes, the mean and standard error of the subtracted money after win/no provocation (blue) or loss/provocation (orange) is shown for the placebo (PL), testosterone + arginine vasopressin (T/AVP).

**Figure 4 brainsci-11-01623-f004:**
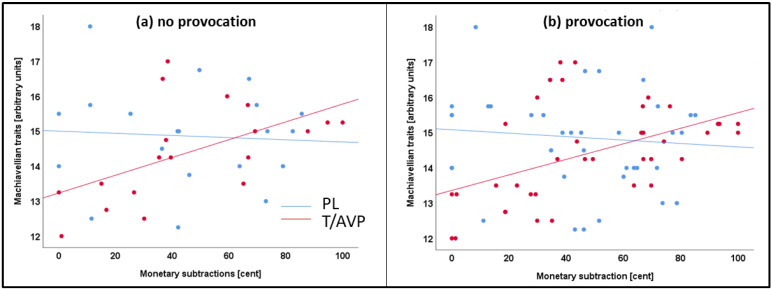
Correlations of monetary subtraction and Machiavellian traits (MACH-IV) are depicted separately for the testosterone + arginine vasopressin (T/AVP) (red) and placebo (PL) (blue) after (**a**) win/no provocation trials and (**b**) loss/provocation trials.

**Figure 5 brainsci-11-01623-f005:**
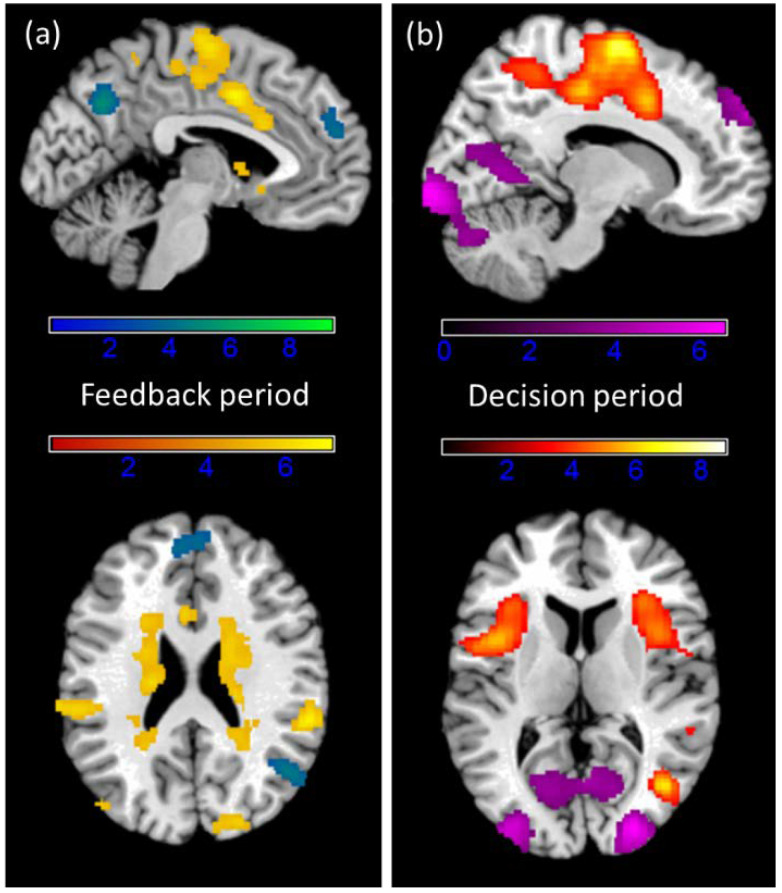
T contrast images in the (**a**) feedback period of win/no provocation > loss/provocation (yellow) and loss/provocation > win/no provocation (blue/green) and in the (**b**) decision period of win > loss (yellow/orange) and loss > win (violet). Clusters are corrected at the voxel level *p* = 0.001 with family-wise error cluster correction.

**Figure 6 brainsci-11-01623-f006:**
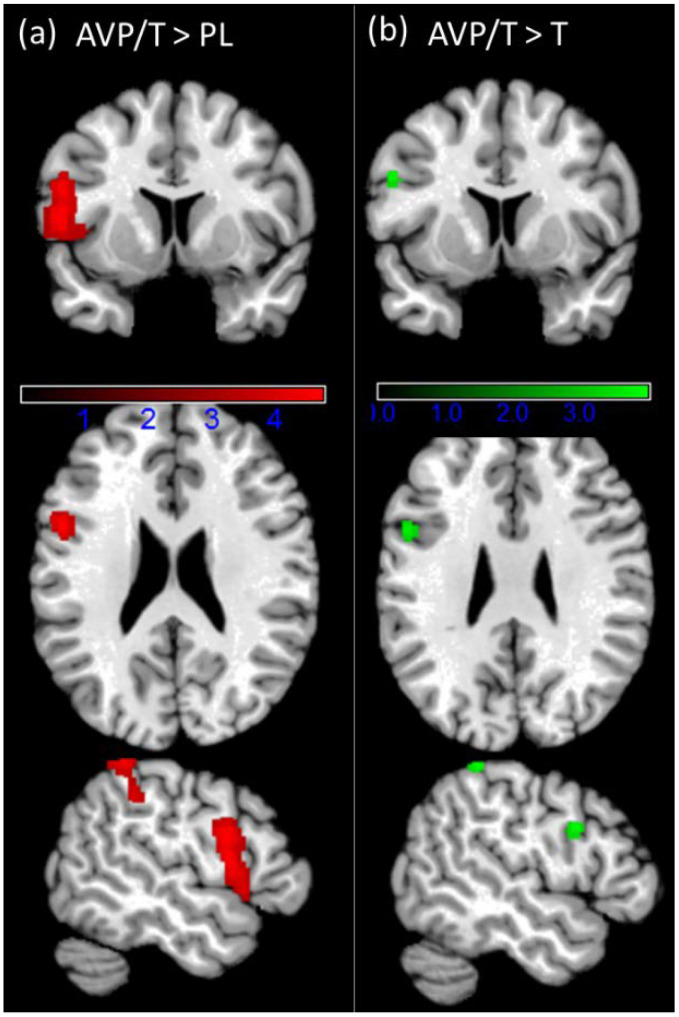
(**a**) Significant clusters of the directed effect T/AVP > PL at the *p* > 0.001 voxel level and FWE *p* < 0.05 cluster-level correction (k > 317) (red)’ and (**b**) significant clusters of the directed effect T/AVP > T (see Appendix A for further details) at the small volume corrected FEW *p* < 0.05.

**Table 1 brainsci-11-01623-t001:** Sample characteristics. Abbreviations: M, mean; SE, standard error of the mean; *p*, *p*-value at significance level of α = 0.05; WST, Wortschatztest (Neuropsychological test for German vocabulary) [46]; BPAQ, Buss Perry Aggression Questionnaire [47]; PPI, Psychopathic Personality Inventory [48]; BIS-11, Barratt Impulsiveness Scale-11 [49]; ERQ, Emotion Regulation Questionnaire [50]; MACH-IV, Machiavellianism Scale [51].

	T/AVP		PL		
	M	SEM	M	SEM	*p*
Age	24.80	1.25	23.95	0.75	0.570
WST	33.05	0.64	33.05	0.63	0.996
BPAQ	62.84	3.04	66.29	3.39	0.455
PPI	343.68	3.46	338.85	2.33	0.249
BIS-11	78.62	1.75	78.87	1.77	0.923
ERQ—reappraisal	25.53	1.17	27.62	1.29	0.241
ERQ—suppression	13.42	0.88	16.33	1.05	0.175
MACH IV	14.42	0.32	14.86	0.32	0.332

**Table 2 brainsci-11-01623-t002:** Statistical effects for the ANOVA testing the influence of provocation (losing versus winning), administration group, and belief about the administration group on monetary subtractions.

Effects	F	*p*	η^2^p
Provocation	2.81	0.102	0.072
Provocation × administration group	**6.44**	**0.016 ***	**0.152**
Provocation × belief	1.05	0.312	0.028
Provocation x administration group × belief	0.10	0.417	0.018
Administration group	0.31	0.581	0.009
Belief	2.34	0.135	0.061
Administration group × belief	0.47	0.496	0.013

Note: * Significant effects are indicated at *p* < 0.05.

**Table 3 brainsci-11-01623-t003:** Peak coordinates (x, y, z) and cluster size (k) for the T contrasts (FWE cluster level corrected) during the feedback period across groups (feedback about winning a trial > feedback about losing and the amount of subtracted money).

Contrast	Region	k	T	x	y	z
	L Insula Lobe	12,210	8.29	−40	0	8
	R Caudate Nucleus		8.15	8	12	−12
	L Putamen		8.13	−14	12	−10
	L Insula Lobe		7.83	−38	4	2
	L IFG (*p*. Opercularis)		7.74	−52	4	6
	R Caudate Nucleus		7.63	16	12	−12
Win > loss	L Middle Temporal Gyrus		7.62	−44	−64	10
	R Putamen		7.42	22	10	2
	R Middle Temporal Gyrus	869	8.84	42	−72	8
	R Cuneus	424	6.89	18	−78	40
	R Superior Occipital Gyrus		6.37	20	−90	34
	R Cuneus		5.66	14	−92	22
	R SupraMarginal Gyrus	382	6.84	58	−38	24
	R Superior Temporal Gyrus		6.73	62	−36	16
	L SupraMarginal Gyrus	376	6.54	−60	−26	34
	L Lingual Gyrus	2632	9.34	−24	−98	−14
	L Inferior Occipital Gyrus		9.19	−36	−90	−12
	L Fusiform Gyrus		8.70	−42	−82	−14
	L Middle Occipital Gyrus		7.25	−32	−96	0
	R Lingual Gyrus	2117	8.21	20	−96	−12
	R Calcarine Gyrus		8.20	18	−102	−6
	R Inferior Occipital Gyrus		8.11	38	−88	−14
	R Middle Temporal Gyrus		3.92	64	−40	−12
	R IFG (*p*. Orbitalis)	1058	5.15	48	26	−14
Loss > win	R Inferior Temporal Gyrus		4.67	48	6	−36
	R Medial Temporal Pole		4.05	44	12	−28
	R Middle Temporal Gyrus		4.02	54	−26	−12
	R IFG (*p*. Orbitalis)		4.01	40	38	−18
	R Medial Temporal Pole		3.97	52	6	−24
	R Superior Medial Gyrus	849	4.92	10	28	62
	L Superior Medial Gyrus		4.26	−6	52	22
	R Superior Parietal Lobule	508	6.02	36	−76	50
	R Angular Gyrus		4.54	50	−64	24
	L Inferior Parietal Lobule	488	4.93	−36	−68	48
	L Angular Gyrus		3.95	−52	−56	28
	L Middle Temporal Gyrus		4.68	−58	−24	−16
	L Precuneus	371	5.78	0	−68	36
	R MCC		3.62	8	−52	34
	L IFG (*p*. Orbitalis)	371	5.57	−30	22	−24
	L Temporal Pole		5.02	−42	20	−24

Note: L = left, R = right.

**Table 4 brainsci-11-01623-t004:** Peak coordinates (x, y, z) and cluster size (k) for the T contrasts (FWE cluster level corrected) during the decision period across groups (decision after won trials > decision after lost trials).

Contrast	Region	k	T	x	y	z
	L Inferior Occipital Gyrus	10,572	9.84	−34	−90	−12
	L Lingual Gyrus		9.28	−22	−100	−14
	R Inferior Occipital Gyrus		9.11	28	−96	−6
	L Middle Occipital Gyrus		8.26	−34	−94	0
	L Inferior Occipital Gyrus		8.06	−48	−68	−16
	R Middle Occipital Gyrus		7.89	28	−90	6
Loss > win	L Calcarine Gyrus	1514	4.74	−22	−66	4
	R Calcarine Gyrus		4.63	4	−72	16
	R Lingual Gyrus		4.29	10	−64	2
	L Cuneus		3.70	−10	−80	16
	R Angular Gyrus		4.91	34	−70	46
	R Middle Occipital Gyrus		4.69	38	−66	34
	L Inferior Parietal Lobule	573	5.20	−34	−60	50
	L Angular Gyrus		3.33	−44	−54	32
	R Inferior Temporal Gyrus	380	5.46	50	−4	−36
	R Medial Temporal Pole		4.76	48	16	−28
	R Temporal Pole		4.70	50	16	−24
	L Paracentral Lobule	331	4.69	−2	−36	80
	L Precuneus		4.44	−4	−40	80
	L IFG (*p*. Triangularis)	274	4.13	−46	14	28
	L Middle Frontal Gyrus		3.46	−46	20	36
	R Superior Medial Gyrus	272	5.17	14	60	32
	L Precentral Gyrus	12,344	10.05	−38	−12	50
	L Posterior−Medial Frontal		8.27	−4	−10	64
	R Posterior−Medial Frontal		8.14	8	−6	66
	L MCC		7.54	−14	−26	44
	R Middle Frontal Gyrus		7.31	42	−4	52
Win > loss	R Precentral Gyrus		7.08	52	0	48
	L MCC		6.86	−6	4	38
	L Insula Lobe	835	5.97	−44	6	6
	L Rolandic Operculum		5.76	−50	4	4
	L IFG (*p*. Triangularis)		4.51	−36	22	8
	R IFG (*p*. Opercularis)	726	5.55	38	12	12
	R Insula Lobe		4.85	32	26	4
	R Rolandic Operculum		3.24	58	0	12
	R Superior Temporal Gyrus	497	5.21	50	−38	22
	R Rolandic Operculum		3.61	40	−30	20
	R Middle Temporal Gyrus	299	6.37	42	−68	8

Note: L = left, R = right.

## Data Availability

The data presented in this study are available on request from the corresponding author. The data are not publicly available due to restrictions in the approved ethics protocol.

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
