# Peer review of "A Combined Administration of Testosterone and Arginine Vasopressin Affects Aggressive Behavior in Males"

_brainsci, 2021, doi:10.3390/brainsci11121623_

Round 1
Reviewer 1 Report
The authors show that a combined administration of testosterone and arginine vasopressin affects aggressive behavior in males. Overall, the manuscript is well-written and the findings are interesting. I only have a minor comment.
The authors mentioned other neurotransmitter systems, such as 5-HT, are involved. Please cite https://doi.org/10.1016/j.neulet.2019.02.022
Author Response
We thank the reviewer for the positive evaluation of our manuscirpt. The suggestion to include a reference for the 5-HT involvement is resonable and we now cite the suggested paper. Please see our changes on page 16: Referring to animal studies, external manipulation of the neurotransmitter systems (e.g., serotonin, GABA) may ultimately activate brain systems related to stressful situations [85] or aggressive behavior [21,86].
85. Hassell, J.E.; Collins, V.E.; Li, H.; Rogers, J.T.; Austin, R.C.; Visceau, C.; Nguyen, K.T.; Orchinik, M.; Lowry, C.A.; Renner, K.J. Local Inhibition of Uptake2 Transporters Augments Stress-Induced Increases in Serotonin in the Rat Central Amygdala. Neurosci. Lett. 2019, 701, 119–124, doi:10.1016/j.neulet.2019.02.022.
Reviewer 2 Report
The manuscript entitled “A combined administration of testosterone and arginine vasopressin affects aggressive behavior in males” is focusing the very interesting field of testosterone- and arginine vasopressin-induced modulation of aggressive behavior. I really enjoyed reading the manuscript and I appreciate your clear and suggestive approach in the evaluation process. The principle question, i.e. the estimation of some specific regional morphofunctional alterations in brain accompanied with induction of aggressive behavior, is relevant and interesting. It originally brings the insight to the site of action for testosterone and arginine vasopressin, using the reliable methodology for both induction and analyses of the outcome, which was not previously estimated. The manuscript is well organized, with clearly presented idea, methodology, and results. The conclusions are within the obtained data allowing better understanding of the complex process evaluated in this study. Still, I have only one suggestion, since it seems reasonable that such a comprehensive study should include a brief mentioning (in the Discussion section) a confirmed role of some other subcortical structures involved in aggressive behavior regulation (such as the hippocampus, some thalamic nuclei, etc.).Author Response
We thank the reviewers for their feedback on our manuscript. We here provide our responses to the reviewers.
It is certainly correct that other brain regions may be involved in the modulation of aggressive behavior. In the current study, we think that social provocation was only a mild stressor, which may explain why we did not observe involvment of suggested structures as the hippocampus. Moreover, neural circuits may not completely overlap across diffrent species, which may also explain why human studies do not replicate findings in rats or other species. However, we agree that it would be useful to extend our discussing on this issue which can be found on page 16:
While in the current study we primarily observed an influence of AVP/ T administra-tion on the left IFG, extending to the left insula and the left IPL, we cannot exclude that the neuromodulators may affect behavioral output by modulating other brain regions on a subthreshold level. A previous study for example, pointed out that activation of AVP1b receptors, which are specifically present in the hippocampal CA2 region, may affect social forms of aggression [82]. In particular, the authors suggest that this recep-tor activation may modulate the hypothalamic-pituitary-adrenal axis under acute stress. In the current study, we may not have observed changes in the activation of the hippocampus because social provocation is only a mild social stressor and may not have led to an acute stress response. Moreover, in rats it has been shown that a vaso-pressin receptor antagonist can suppress the distributed neural circuit involved in ag-gressive motivation by specifically activating the anterior thalamic nuclei [83]. It cur-rently remains unclear if such findings may be translated to humans mostly due to differences in the definition of the neural circuits, which in humans may include a broad involvement of prefrontal brain regions [84].
82. Caldwell, H.K.; Aulino, E.A.; Rodriguez, K.M.; Witchey, S.K.; Yaw, A.M. Social Context, Stress, Neuropsychiatric Disorders, and the Vasopressin 1b Receptor. Front. Neurosci. 2017, 11, 567, doi:10.3389/fnins.2017.00567.
83. Ferris, C.F.; Stolberg, T.; Kulkarni, P.; Murugavel, M.; Blanchard, R.; Blanchard, D.C.; Febo, M.; Brevard, M.; Simon, N.G. Imaging the Neural Circuitry and Chemical Control of Aggressive Motivation. BMC Neurosci. 2008, 9, 111, doi:10.1186/1471-2202-9-111.
84. Repple, J.; Pawliczek, C.M.; Voss, B.; Siegel, S.; Schneider, F.; Kohn, N.; Habel, U. From Provocation to Aggression: The Neural Network. BMC Neurosci. 2017, 18, 73, doi:10.1186/s12868-017-0390-z.